# Herbivore-shrub interactions influence ecosystem respiration and BVOC composition in the subarctic

Cole G. Brachmann[1,2], Tage Vowles[2,3], Riikka Rinnan[4,5], Mats P. Björkman[1,2], Anna Ekberg[6], and Robert G. Björk[1,2]

[1]Department of Earth Sciences, University of Gothenburg, Gothenburg, Sweden

[2]Gothenburg Global Biodiversity Centre, Gothenburg, Sweden

[3]IVL Swedish Environmental Research Institute, Gothenburg, Sweden

[4]Center for Volatile Interactions (VOLT), Department of Biology, University of Copenhagen, Copenhagen, Denmark

[5]Center for Permafrost (CENPERM), Department of Geosciences and Natural Resource Management, University of Copenhagen, Copenhagen, Denmark

[6]Centre for Environmental and Climate Science, Lund University, Lund, Sweden

*Correspondence to*: Cole G. Brachmann (cole.brachmann@gu.se)

**Abstract.** Arctic ecosystems are warming nearly four times faster than the global average which is resulting in plant community shifts and subsequent changes in biogeochemical processes such as gaseous fluxes. Additionally, herbivores shape plant communities and thereby they may alter the magnitude and composition of ecosystem respiration and BVOC emissions. Here we determine the effect of large mammalian herbivores on ecosystem respiration and BVOC emissions in two southern and two northern sites in Swedish Scandes, encompassing mountain birch (LOMB) and shrub heath (LORI) communities in the south and low-herb meadow (RIGA) and shrub heath (RIRI) communities in the north. Herbivory significantly altered BVOC composition between sites and decreased ecosystem respiration at RIGA. The difference in graminoid cover was found to have a large effect on ecosystem respiration between sites as RIGA, with the highest cover, had 35% higher emissions than the next highest emitting site (LOMB). Additionally, LOMB had the highest emissions of terpenes with the northern sites having significantly lower emissions. Differences between sites were primarily due to differences in exclosure effects, soil temperature and prevalence of different shrub growth forms. Our results suggest that herbivory has a significant effect on trace gas fluxes in a productive meadow community and differences between communities may be driven by differences in shrub composition.

## 1 Introduction

Arctic ecosystems are particularly susceptible to climate change effects as the rate of warming is nearly four times that of temperate regions (Rantanen et al., 2022). This leads to climate driven vegetation shifts in tundra environments that will have associated repercussions on soil greenhouse gas (GHG), and other trace gas, fluxes (Virkkala et al., 2018), which are important environmental properties as they feedback directly to the climate. Plant communities largely determine the production and consumption of trace gases through photosynthesis and respiration, production of secondary compounds (such as biogenic volatile organic compounds: BVOCs), and regulation of the microbial community (Ward et al., 2013). The plant species composition of a community then influences the magnitude of the trace gas fluxes (such as $CO_2$) and the composition and quantity of the BVOC species released.

BVOCs are chemicals produced by plants for a variety of purposes including reproductive signalling, communication, and herbivore deterrence (Peñuelas and Staudt, 2010). Their emissions are largely determined by plant community structure in terms of magnitude and composition, although their emissions are predicted to increase as a direct consequence of higher temperatures in high latitude communities (Rinnan et al., 2020). BVOCs play a role in climate warming primarily through their interaction with ozone, effects on the lifetime of methane in the atmosphere, and the formation of secondary organic aerosols (Boy et al., 2022; Calfapietra et al., 2013; Peñuelas and Staudt, 2010). Secondary organic aerosols may have an overall cooling effect by scattering light and leading to cloud formation (Shrivastava et al., 2017; Spracklen et al., 2008); therefore, understanding the emissions and identity of BVOC emissions is critical to future climate projections as they can act to both enhance or mediate climate change effects.

$CO_2$ is the primary GHG responsible for climate warming, and its release (ecosystem respiration; ER) from plants and soils in the Arctic is crucial, as Arctic soils store approximately 50% of the global terrestrial belowground carbon (Mishra et al., 2021; Hugelius et al., 2014, 2013; Sistla et al., 2013). ER is one of the largest contributors to $CO_2$ emissions globally (Liu et al., 2022; Sharkhuu et al., 2016) and is predicted to be increasingly vital to the $CO_2$ balance of northern regions as it is driven by vegetation change, productivity and microbial activity (Liu et al., 2022; Parker et al., 2015). Thus, influences of vegetation changes on BVOC emissions and ER can impact the feedback effects on climate change, potentially exacerbating or mediating its effects, with implications regionally and globally (Peñuelas and Staudt, 2010; Heimann and Reichstein, 2008).

Increases in vegetation biomass in tundra could increase the magnitude of BVOC fluxes (Rinnan et al., 2011), however, vegetation composition changes will likely have a stronger effect on the composition of BVOCs emitted as the composition of VOCs emitted is plant species-specific (Peñuelas and Staudt, 2010). Some general patterns

can be drawn between certain types of BVOCs and plant functional groupings, such as isoprene emissions
increasing with a higher abundance of sedges or willows and monoterpenes becoming more prevalent with
increasing evergreen dwarf shrubs or birch (Männistö et al., 2023; Rinnan et al., 2020; Svendsen et al., 2016).
Furthermore, the magnitude and composition of BVOCs emitted from soil can be influenced by the identity and
quality of plant litter (Svendsen et al., 2018). Therefore, changes in the vegetation community are likely to have
large impacts on BVOC emissions.

The magnitude of ER also differs between plant communities within the forest-tundra ecotone (Treat et al., 2018).
Subarctic birch forest, composed of mountain birch trees and deciduous shrubs primarily, have larger ER
compared to tundra heath and meadow communities due to the high contribution from woody deciduous
vegetation and stimulation of microbial communities through litter and mycorrhizal inputs to the soil (Virkkala et
al., 2021; Strimbeck et al., 2019; Parker et al., 2015). Heath communities have been found to have the lowest ER
from these community types which is attributed to the relatively slow decomposition of organic matter in the soil
(Sørensen et al., 2018; Parker et al., 2015). Concomitantly, the shift to ectomycorrhiza-dominated communities
coincides with increased ER as they more effectively scavenge organic carbon and contribute to higher
productivity (Parker et al., 2015). Shifts in plant community composition due to climate change have been well
documented in tundra ecosystems (Bjorkman et al., 2020; Elmendorf et al., 2012), and have subsequent effects
on BVOC emissions and ER (Virkkala et al., 2018; Valolahti et al., 2015).

Herbivores can act to either enhance or mediate changes in vegetation communities through selective foraging
and nutrient input (Ylänne et al., 2020; Barthelemy et al., 2018; Vowles et al., 2017b; Olofsson et al., 2009). Both
selective foraging and nutrient addition can have large impacts on the plant community by shifting the competitive
advantage of certain species and stimulating microbial activity (Bardgett and Wardle, 2003; Olofsson et al., 2002).
Herbivory likely influences soil gaseous fluxes indirectly through altering the trajectory of vegetation community
changes (Metcalfe and Olofsson, 2015; Cahoon et al., 2012). Therefore, by changing the plant community in an
area, herbivores can influence the magnitude and direction of both BVOC fluxes and ER (Vowles and Björk,
2019). However, studies assessing the consequences of large herbivore grazing on BVOC fluxes and ER are
lacking in tundra ecosystems and are important for a broader understanding of feedback mechanisms in the Arctic
(Vowles and Björk, 2019; Köster et al., 2018; Ylänne et al., 2015; Metcalfe and Olofsson, 2015; Cahoon et al.,
2012).

Thus, here we investigate the role of herbivory in regulating BVOC emissions and ER through interactions with
vegetation communities in Swedish montane and tundra habitats. If herbivory is a driver of these fluxes, we
predicted that excluding herbivores would increase ER and alter BVOC composition by shifting the plant

community to one more dominated by deciduous shrubs, as has been identified at these sites previously (Vowles et al., 2017 b, a). Therefore, we hypothesize that differences between plant communities, in terms of dominance of functional types, soil characteristics and climate properties, would be the strongest predictor of trace gas fluxes, regardless of the presence of herbivory. This paper aims to address two main questions: (1) How do vegetation shifts indirectly caused by large herbivore exclusion affect the magnitude of ER and composition of BVOC fluxes? (2) Are these effects consistent among different vegetation communities?

## 2 Methods

### 2.1 Study sites

The study was conducted at four sites within the Swedish Scandes, two in the south and two in the north of the mountain range, from June 2013 to June 2014 (Fig. 1). These sites encompassed three distinct community types: birch forest, shrub heath, and low-herb meadow. The southern sites included the birch forest and a shrub heath community, and the northern sites contained the low herb meadow and a second shrub heath community. The southern sites are referred to as: Långfjället mountain birch forest (LOMB hereafter; 62°03'59"N, 12°14'56"E; 809 m a.s.l.) and Långfjället shrub heath (LORI hereafter; 62°06'53"N, 12°16'30"E; 853 m a.s.l.), located approximately 5 km apart near lake Grövelsjön in the county of Dalarna. The Northern sites are referred to as: Ritsem shrub heath (RIRI hereafter; 67°46'33"N, 17°32'22"E; 847 m a.s.l.) and Ritsem low herb meadow (RIGA hereafter; 67°49'35"N, 17°43'02"E; 719 m a.s.l.), located approximately 10 km apart near Ritsem, Norrbotten County. The primary large mammalian herbivore in our sites is reindeer (*Rangifer tarandus tarandus*) for which densities of 2.8 reindeer per $km^2$ near LOMB and LORI, 2.2 reindeer per $km^2$ near RIGA and 1.4 reindeer per $km^2$ near RIRI were previously reported for the three Sami herding villages nearest our sites (Vowles et al., 2017a, b). Moose (*Alces alces*) are another large herbivore that could be present at all study sites, with potential instances of roe deer (*Capreolus capreolus*) at LOMB. The moose populations are similar for the management areas in which our sites are located, and both have a density of approximately $0.1 – 0.2$ moose per $km^2$ over the study period according to county board hunting statistics (SCAB Statistik Älgdata accessed 2022).

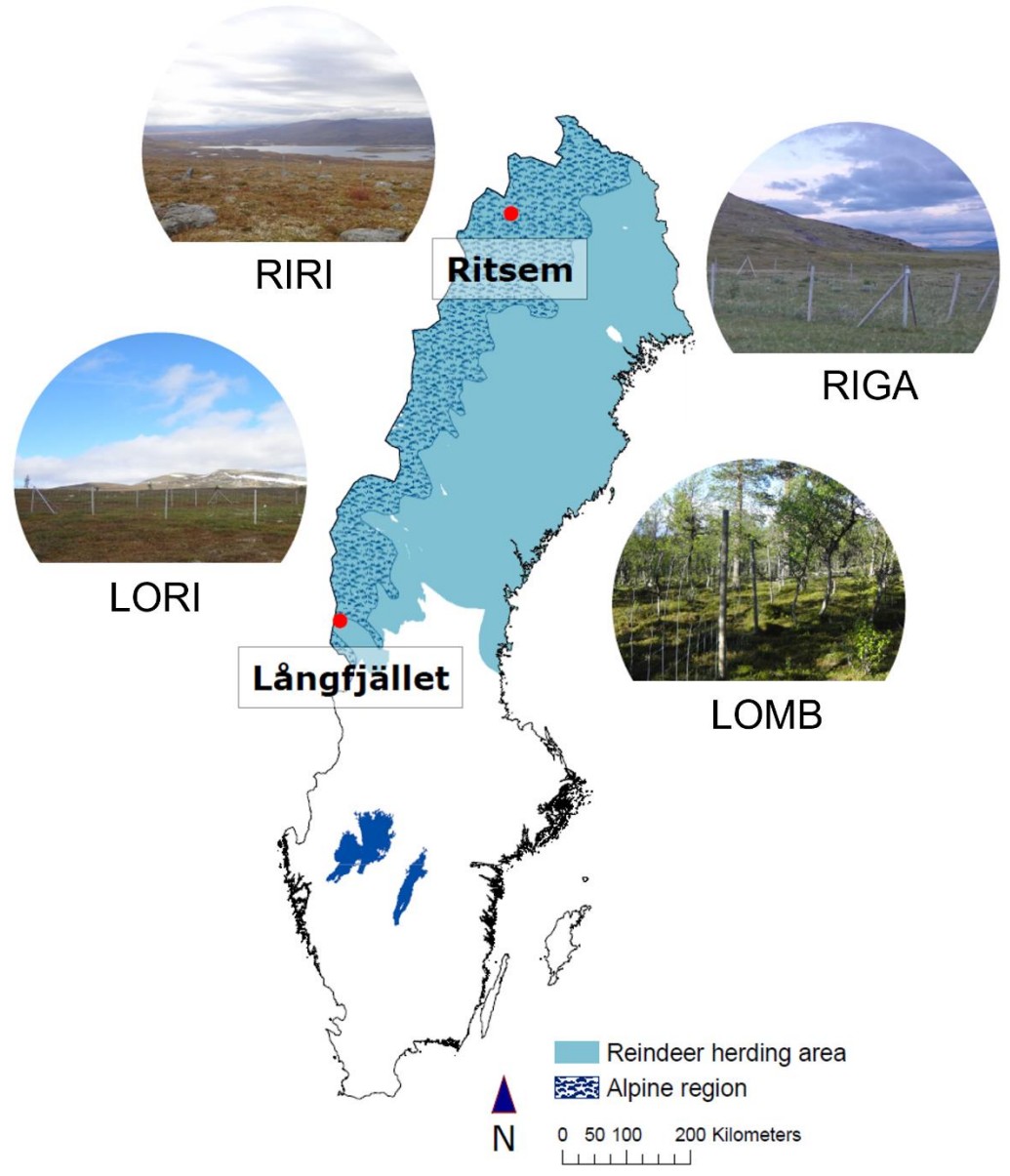

 **Fig. 1. Map of study sites and communities with overlay of reindeer herding area and alpine regions in Sweden. Photo credits: Tage Vowles.**

The vegetation communities at each of the four sites have been previously described in detail (Vowles et al., 2017a, b); and the climate properties are reported in Table 1. LOMB contains a tree layer almost entirely of *Betula*

*pubescens ssp. tortuosa*, with a field layer containing dwarf shrubs: *Empetrum nigrum*, *Vaccinium myrtillus* and *Vaccinium vitis-idaea*, graminoids: *Deschampsia flexuosa* and *Nardus stricta*, and forbs: *Melanpyrum pratense* and *Solidago virgaurea*. LORI is composed mostly of dwarf shrubs, mainly: *E. nigrum*, *V. myrtillus*, *V. vitis-idaea*, *Calluna vulgaris* and *Betula nana*. This site also contains some limited graminoids, primarily *D. flexuosa*. RIRI contains the same species as previously mentioned plus additional graminoid and forb species as the bedrock is calcareous providing more nutrients into the soil. The predominant additional graminoids are *Calamagrostis lapponica* and *Carex bigelowii*, and the additional forbs are *Bistorta vivipara* and *Hieracium sect. Alpina*. Finally, RIGA is primarily composed of shrubs: *B. nana*, and Salix sp., graminoids: *Deschampsia cespitosa*, *D. flexuosa*, and *Carex aquatilis*, and forbs: *Viola biflora*, *Thalictrum alpinum* and *Saussurea alpina*.

**Table 1. Site characteristics for each location used in the study. Mean annual air temperature (MAAT), mean annual soil temperature (MAST) and soil moisture are reported for the duration of the study (June 2013 - June 2014). Mean annual precipitation (MAP) is reported from 1961-1990 from the nearest SMHI weather station to each site, which is Grövelsjön for LOMB and LORI, and Ritsem for RIGA and RIRI. NH4+ and NO3- are reported separately for summer and winter PRS probe burial periods. The burial length differs slightly between sites and seasons, with LOMB and LORI buried for 121 days in summer and 235 / 236 days in winter, respectively. RIGA and RIRI were buried for 71 days in summer and 283 days in winter. The LORI3 plot was omitted as an outlier for all means.**

| Site | MAAT (°C) | MAP (mm) | MAST (°C) | Soil Moisture (%) | Summer $NH_4^+$ (µg 10 $cm^{-2}$ burial length$^{-1}$) | Summer $NO_3^-$ (µg 10 $cm^{-2}$ burial length$^{-1}$) | Winter $NH_4^+$ (µg 10 $cm^{-2}$ burial length$^{-1}$) | Winter $NO_3^-$ (µg 10 $cm^{-2}$ burial length$^{-1}$) |
|---|---|---|---|---|---|---|---|---|
| LOMB | 2.8 | 697 | 11.5 | 19.9 | 14.6 | 3.5 | 17.1 | 3.6 |
| LORI | 2.8 | 697 | 11.6 | 28.9 | 10.6 | 1.9 | 21.9 | 2.2 |
| RIGA | 0.9 | 510 | 11.3 | 78.7 | 4.0 | 3.3 | 5.3 | 5.9 |
| RIRI | -0.3 | 510 | 10.8 | 38.6 | 7.3 | 1.9 | 5.6 | 2.4 |

The effect of herbivory on ER and BVOC fluxes was determined using herbivore exclosure fences. Three exclosure plots and three paired ambient plots (25 x 25 m) were installed at each site in 1995 and are composed of wire mesh 1.7 m high that functions to exclude reindeer and other large mammalian herbivores from accessing the sites (Vowles et al., 2017a, b). By preventing access to a site, exclosure fences allow for an estimation of the effect of large mammal herbivory by comparing to ambient conditions in pairwise 25 x 25 m plots without fences. Each site has three paired exclosure plots and ambient plots, however, in RIGA only two of the three originally established ambient plots could be located and so a new ambient plot was established in 2012 to keep the design balanced (Vowles et al., 2017a).

## 2.2 BVOC emissions

BVOC fluxes were measured twice for each site during the growing season, in early July and again in late July/ early August (Table A1). Three permanent PVC soil collars (10 cm diameter) were inserted at random locations within the central area of each plot, at least 1 m from the edge a day prior to the first measurement. BVOC fluxes were then measured using transparent teflon chambers fitted onto these soil collars at the time of measurement with a temperature logger connected to the chamber to record temperature throughout the measurement interval. A pump was used to circulate air from the chamber through stainless steel adsorbent cartridges containing 150 mg Tenax TA and 200 mg Carbograph 1TD (Markes International Limited) at 200 ml min-1 and then back into the chamber for a through-flow measurement of BVOCs over 20 minutes. At the end of the measurement, the collected air sample volume was recorded to calculate the BVOC flux. The adsorbent cartridges were analyzed using gas chromatography-mass spectrometry following thermal desorption (Clarus 500, PerkinElmer, Waltham, MA, USA; Ekberg et al., 2009). The obtained chromatograms were analyzed using PARADISe software (Johnsen et al., 2017) and the compounds identified by matching with the NIST mass spectral library. Terpene compounds were quantified by comparing to standards where pure standards were used for identification and quantification of α-pinene, β-pinene, 3-carene, limonene, eucalyptol and caryophyllene, while for all other monoterpenes and sesquiterpenes, α -pinene and caryophyllene were used for quantification, respectively. Mean terpene emissions are detailed for each site and treatment in Table S2. BVOC emission rates were calculated for monoterpenes (MT) and sesquiterpenes (SQT), while the NIST-identified dataset with peak areas of all other compounds was used to describe the chemical composition of the emitted BVOC blend.

## 2.3 Ecosystem respiration

ER during the growing season was measured using a closed-chamber technique (Björkman et al., 2010a). An opaque chamber was sealed onto the collar during measurements where air from the headspace was circulated into 20 ml sample vials over 30 s using an electric pump (flow rate 0.5L/min). Samples were obtained at 3, 6, 10, 30, and 50 minutes after the chamber was sealed onto the collar. The samples were analyzed for $CO_2$ concentration using gas chromatography (Agilent 7890A GC coupled to an Isoprime GC 5 interface and an Isoprime 100 IRMS, Aglient Technologies, Santa Clara, U.S.) and fluxes estimated as a linear change in $CO_2$ concentration over time. Growing season fluxes were measured from late June – early October in the southern sites and late June – early September in the northern sites, and again in early June the following year for all sites.

Winter ER was estimated during the snow-covered period at the LORI and LOMB sites based on Fick´s first law on diffusion (Pirk et al., 2016; Björkman et al., 2010b; Sommerfeld et al., 1993). Air samples were withdrawn

from the snowpack (at every 10 cm) using a gas-tight syringe fitted to 1/6" stainless steel tubings attached to an

       avalanche probe inserted into the snow above each flux collar. The air samples were then transferred to headspace

       vials for storage until analysed by gas chromatography. After air sampling, snow density, snow temperature and

       snow profile characteristics were collected from adjacent snowpack (to avoid disruption of the snowpack at the

       sampling location) to be used in the flux calculations (see Björkman et al., 2010b for further details).

$Q_{10}$-values for each of the collars for the growing season (RIRI and RIGA) and for the full year (LORI and LOMB)

       were estimated based on the Arrhenius equation, by plotting the natural logarithm of the $CO_2$ emissions against

       the measured soil temperature (in 1000/K) as outlined in Davidson and Janssens (2006). Furthermore, to enable a

       direct comparison between the sites, an interpolation approach (Björkman et al., 2010a) was used for the growing

       season data where data was first interpolated between two conjuncting measurements to generate a flux per day

and summed up as cumulative count of emissions during July 02 - September 02, 2013.

**2.4 Vegetation assessments**

The vegetation was measured in each plot using twenty 1 m$^2$ subplots within which cover of each species was

visually estimated (Table 2; Vowles et al., 2017b). All subplots were located within a 22 x 22 m area to reduce

edge effects from influencing the estimations. The total cover estimates were allowed to go beyond 100% as

individuals may go beyond the range of the subplot and to account for overlapping layers of vegetation. All

identified species were grouped into growth form categories to use in an ordination analysis. The growth forms

are deciduous prostrate dwarf shrub, deciduous semi-prostrate dwarf shrub, deciduous tall shrub, evergreen

prostrate dwarf shrub, evergreen semi-prostrate dwarf shrub, evergreen tall shrub, forb, graminoid, non-vascular

species, and other which encompasses the percent ground cover attributable to abiotic and bare ground

components. The vegetation within the chamber is not necessarily 1:1 with the surrounding vegetation surveyed,

but we proceed on the assumption that the chamber takes a representative subsample of the vegetation or that the

measured fluxes can be tied to the prevalence of vegetation types in the exclosure/ambient plot.

**Table 2. Mean and standard error of % cover of plant functional types across each community and treatment condition.**

| | RIRI | | RIGA | | LORI | | LOMB | |
|---|---|---|---|---|---|---|---|---|
| | Exclosure | Ambient | Exclosure | Ambient | Exclosure | Ambient | Exclosure | Ambient |
| Deciduous tall shrub | 9.9 (9.1) | 1.9 (1.1) | 2.5 (1.3) | 1.2 (0.3) | 19.3 (3.8) | 10.7 (4.3) | 6.3 (0.6) | 6.7 (2.2) |
| Deciduous semi-prostrate dwarf shrub | 6.6 (3.8) | 6.8 (4.4) | 0.0 (0.0) | 0.0 (0.0) | 2.4 (1.0) | 1.6 (0.7) | 12.6 (3.2) | 10.3 (0.2) |
| Deciduous prostrate dwarf shrub | 25.7 (8.7) | 11.2 (3.8) | 1.2 (0.8) | 1.9 (0.9) | 0.1 (0.1) | 0.0 (0.0) | 0.0 (0.0) | 0.0 (0.0) |
| Evergreen tall shrub | 0.0 (0.0) | 0.0 (0.0) | 0.0 (0.0) | 0.0 (0.0) | 0.0 (0.0) | 0.0 (0.0) | 2.3 (0.9) | 2.3 (1.1) |
| Evergreen semi-prostrate dwarf shrub | 10.8 (5.2) | 9.7 (4.1) | 0.3 (0.2) | 5.1 (3.9) | 84.0 (5.3) | 93.1 (4.3) | 53.2 (4.2) | 38.5 (14.5) |
| Evergreen prostrate dwarf shrub | 0.8 (0.3) | 1.1 (0.6) | 0.0 (0.0) | 0.0 (0.0) | 0.3 (0.2) | 0.0 (0.0) | 0.0 (0.0) | 0.0 (0.0) |
| Graminoid | 6.8 (0.7) | 21.3 (10.7) | 44.4 (9.6) | 48.1 (4.6) | 1.8 (0.3) | 1.6 (0.4) | 19.8 (1.4) | 42.5 (22.4) |
| Forb | 6.8 (0.7) | 10.3 (2.0) | 33.1 (2.2) | 16.2 (0.7) | 0.0 (0.0) | 0.1 (0.1) | 1.8 (0.4) | 4.8 (2.9) |
| Non-vascular | 44.7 (2.6) | 46.5 (3.5) | 37.7 (9.9) | 44.6 (5.6) | 40.1 (4.6) | 33.8 (2.9) | 53.4 (4.1) | 43.6 (8.7) |
| Abiotic | 10.8 (3.6) | 9.1 (3.4) | 0.0 (0.0) | 0.0 (0.0) | 0.4 (0.4) | 1.9 (1.0) | 1.2 (1.1) | 1.2 (0.4) |
| Total | 122.8 (5.3) | 118.0 (3.4) | 119.2 (4.5) | 117.1 (5.4) | 148.4 (10.1) | 142.9 (4.2) | 150.6 (5.3) | 149.9 (7.6) |

### 2.5 Abiotic conditions

Temperature loggers (Tinytag plus 2 TGP-4020; Gemini Data Loggers, Chichester, UK) were placed in the centre of each plot which measured hourly soil temperatures at 2 cm depth for the duration of the experiment. From the temperature data, thawing degree-days (TDD), which is the sum of all mean daily temperatures above 0°C, were calculated from the soil temperature data according to Molau and Mølgaard (1996) for the period that the chambers were in the ground.

Air temperature was also recorded hourly by one logger (Tinytag plus 2 TGP-4500; Gemini Data Loggers, Chichester, UK) at each site, at a height of approximately 2 m (Table S3). Mean temperatures were calculated from the loggers at each site for the experimental period. In order to obtain a mean temperature for a whole year, site means were calculated from June 12, 2013 – June 11, 2014 at the Långfjället sites and from June 27, 2013 – June 26, 2014 at the Ritsem sites. Minor gaps (no more than seven days) in the temperature series caused by

malfunctioning loggers were filled in using linear regression against the logger which gave the highest $R^2$-value. Soil moisture was measured from the top 6 cm on the same sampling dates as ER using a Delta ML2x Theta probe (Delta-T Devices Ltd, Cambridge, U.K.). Moisture was measured as % water content in the soil (Fig. A1). Plant Root Simulator (PRS®) Probes (Western Ag Innovations, Inc., Saskatoon, Canada), which contain ion exchange resin membranes, were used to measure soil $NO_3^-$ and $NH_4^+$ availability in situ at each plot. Four cation

and four anion probes were installed to 10 cm depth, close to the centre of each plot, at the beginning of the experimental period. Before the winter season, the original probes were removed and replaced with a new set of probes, which were then removed at the end of the experiment. After removal, the probes were cleaned and sent to Western Ag Innovations in Saskatoon, Canada, for ion extraction and analysis.

### 2.6 Statistical analyses

All statistical analyses were carried out with R statistical software version 4.2.1 (R Core Team, 2022). To determine the direct effect of site and herbivory on terpene emissions the differences in MT and SQT emissions from each site and treatment (herbivory present in ambient plots or absent in exclosure plots) were evaluated using linear mixed effects models using the lme4 package (Bates et al., 2015) on log transformed data. The best performing model for MT data included site and treatment as fixed effects with plot and date as random effects,

and the best performing model for SQT data included site, treatment and soil temperature as fixed effects with plot and date included as random effects. A linear mixed model was used to follow up on β-pinene differences using site and treatment as fixed effects, and plot and date as random effects. To investigate how site and herbivory condition effected BVOC composition, the differences in BVOC composition from each measurement was

evaluated using a redundancy analysis (RDA) ordination using the vegan package (Oksanen et al., 2019) on Hellinger transformed total BVOC compound data. As a constrained ordination, RDA also allows for determination of the relationships between the measurements and environmental variables (treatment, soil temperature, percent cover of vegetation growth forms in the surrounding area, $NH_4^+$ and $NO_3^-$) which were confirmed through an accompanying ANOVA on the RDA output. The interaction between herbivory and the shrub types was also evaluated through RDA and subsequent ANOVA. 85% Confidence ellipses were drawn for each group within the RDA as they have been shown to have a good fit with data without being too conservative as estimates (Payton et al., 2003, 2000).

The evaluate the effect of site and herbivory condition on ER, a mixed effects model was used to explore differences in ER between exclosure and ambient plots, with treatment, site, date, soil temperature, percent cover of graminoids, and site:treatment and soil temp:graminoids interactions as fixed factors and plot as random factor using the nlme package (Pinheiro et al., 2015). Graminoids were the only vegetation type included as they were the only that showed a significant effect on ER, though all others measured were tested with the model. Soil moisture was also tested but left out of the final model after stepwise reduction. Additionally, a rational quadratic correlation structure was included in the model to account for temporal correlation between measurements (Kravchenko and Robertson, 2015). Several correlation structures were tested (including autoregressive, linear, exponential, Gaussian and spherical) and the Akaike information criterion was used to select the best fitting structure. The emmeans package (Lenth, 2023) was used to test for treatment effects at individual sites, using pairwise t-tests with Bonferroni P-value adjustments. An additional mixed effects model was constructed to evaluate treatment differences within RIGA specifically to determine if vegetation or soil temperature drove the treatment effect observed. Multiple models were evaluated to determine if any plant functional type could drive the treatment effect and the final model contained treatment, date, soil temperature and percent cover of evergreen prostrate dwarf shrubs as fixed effects with plot as a random effect and a rational quadratic correlation structure. Model parameters for both the overall and RIGA-only models were evaluated with a mixed model analysis of variance (ANOVA) for significance. All analyses utilized an alpha of 0.05 for significance where appropriate.

## 3 Results

### 3.1 BVOC emissions

Herbivory did not have significant effects on MT or SQT emission rates in any of the plant communities (Fig. 2). However, for MT emissions RIRI ($T_{4.09}$ = -3.48, p = 0.024) was significantly different from LOMB ambient (as

reference level); subsequently the intercept (reference level in this context) is also significant ($T_{3.96} = 17.57$, p < 0.001). SQT emissions were not different between any of the sites, however the intercept (ambient LOMB with a temperature of 0 °C) was significant ($T_{11.17} = 3.02$, p = 0.012) indicating that soil temperature may influence SQT emissions but over a larger range than we had sampled.

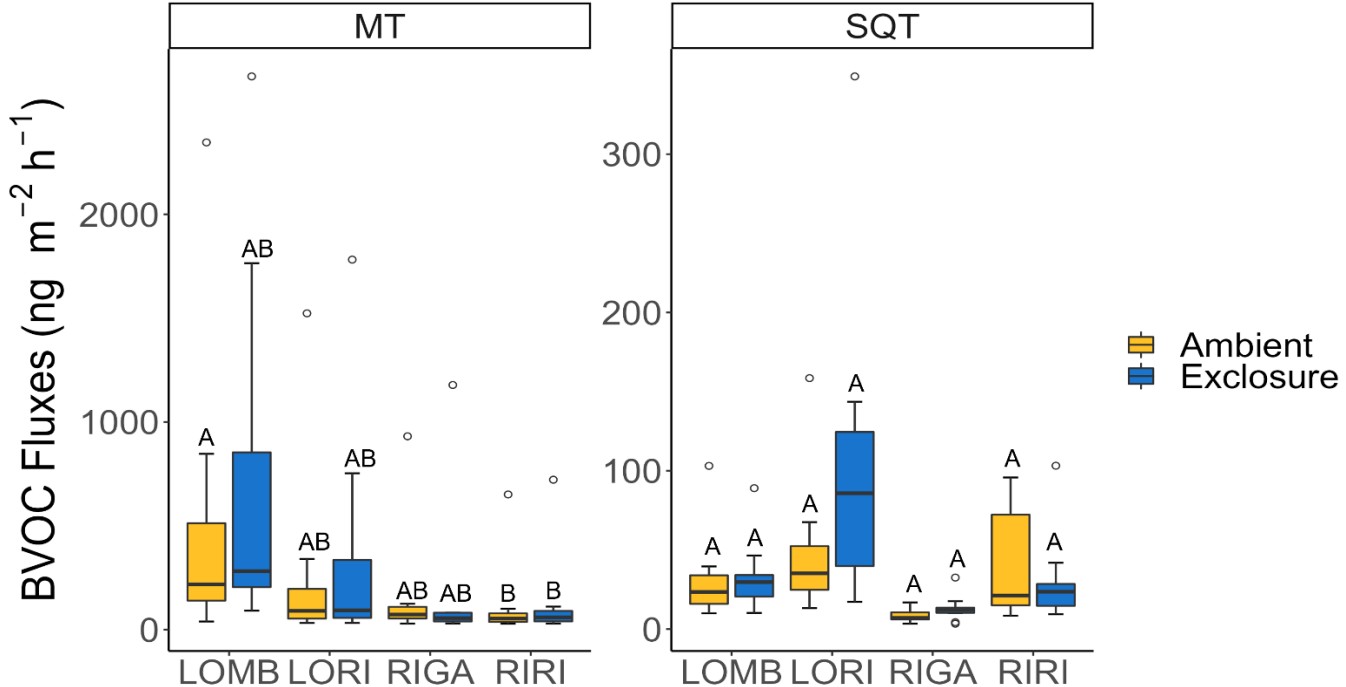

**Fig. 2. Log of average monoterpene (MT) and sesquiterpene (SQT) emissions from each site in each treatment condition. Herbivory did not affect the magnitude of either MT or SQT emission at any site. Letters denote significant differences between sites and treatment according to linear mixed models on log transformed data and are assigned independently for each panel.**

The redundancy analysis showed clear distinctions between the four sites in their emitted BVOC composition (Fig. 3), although the 85% confidence ellipses overlapped between all sites. The analysis determined that treatment ($F_{1,79} = 3.427$, p = 0.005), soil temperature ($F_{1,79} = 13.822$, p < 0.001), percent cover of abiotic components ($F_{1,79} = 2.292$, p = 0.042) and all shrub categories (deciduous prostrate dwarf shrub: $F_{1,79} = 9.162$, p < 0.001, deciduous semi-prostrate dwarf shrub: $F_{1,79} = 4.973$, p < 0.001, deciduous tall shrub: $F_{1,79} = 6.841$, p < 0.001, evergreen prostrate dwarf shrub: $F_{1,79} = 5.211$, p < 0.001, evergreen semi-prostrate dwarf shrub: $F_{1,79} = 6.663$, p < 0.001,

evergreen tall shrub: $F_{1,79} = 2.548$, $p = 0.015$) had significant effects on the differences between measurements in terms of their BVOC compositions. The treatment vector in Figure 3 points towards the exclosure condition where herbivores are excluded. Additionally, of the interactions between treatment and shrub types that were investigated, only deciduous prostrate shrubs were significant drivers of BVOC composition ($F_{1,79} = 4.149$, $p < 0.001$). Two BVOCs had correlations greater than 0.7 and are important for distinguishing between sites (2-Ethylfuran: $r = 0.71$, $p = 0.01$; and β-pinene: $r = 0.75$, $p = 0.01$); although, 2-Ethylfuran is a tentative NIST identification, whereas β-pinene was identified by comparing to a standard and so the confidence of its assignment is high. β-pinene emissions were marginally lower in RIRI ($T_{3.96} = -2.67$, $p = 0.055$) as compared to LOMB (as the reference level in the model) but the intercept itself was significant ($T_{4.12} = 21.09$, $p < 0.001$), however treatment did not have an effect on β-pinene emissions.

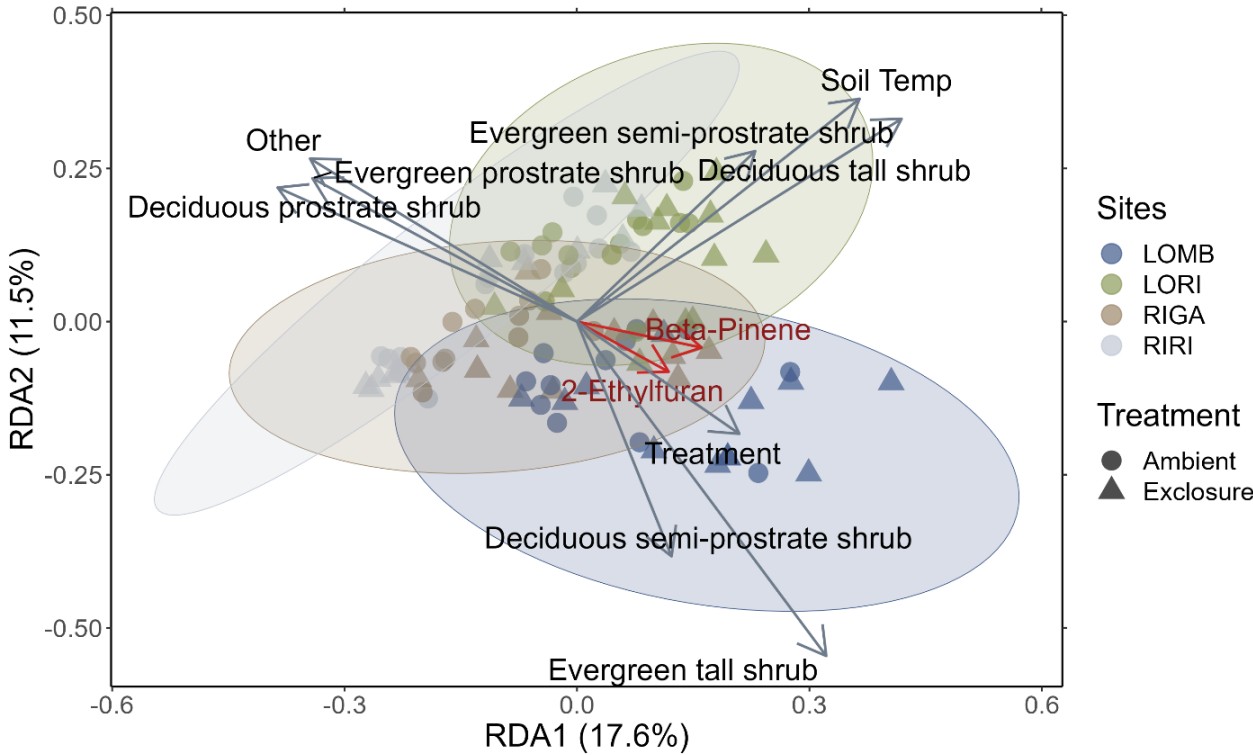

**Fig. 3. Redundancy analysis of BVOC compounds emitted from each site constrained by environmental properties. Circles represent ambient and triangles exclosure plots. Vectors correspond to significant environmental variables with the length of the vector representing the strength of the relationship. Ellipses are 85% confidence ellipses that**

**correspond to the standard deviation of the plotted points for each site separately. The red vectors correspond to the BVOC compounds with correlation values greater than 0.7. Treatment, soil temperature, percent cover of abiotic components and all shrub categories were significantly related to the composition of BVOCs emitted from each measurement (sites scores).**

### 3.2 Ecosystem respiration

Growing season respiration varied between sites, with RIGA having the highest average ER ($215\pm19$ mg $CO_2$ m$^{-2}$ h$^{-1}$), which was 35% higher than LOMB ($159\pm17$ mg $CO_2$ m$^{-2}$ h$^{-1}$), and about three times higher than both shrub heath sites ($83\pm7$ and $58\pm8$ mg $CO_2$ m$^{-2}$ h$^{-1}$; at Ritsem and Långfjället respectively). Significant growing-season Arrhenius relationships were only found at the RIGA ($Q_{10} = 3.0\pm0.2$ and $2.8\pm1.0$, exclosure and ambient plots respectively) and RIRI ($Q_{10} = 4.4\pm2.7$ and $11.3\pm4.2$, exclosure and ambient plots respectively) sites (Fig. 4). Annual Arrhenius relationships were found to be significant for most LOMB ($Q_{10} = 33.1\pm17.5$ and $18.0\pm9.4$, exclosure and ambient plots respectively) and LORI ($Q_{10} = 8.0\pm3.6$ and $5.5\pm1.1$, exclosure and ambient plots respectively) plots, although the predictiveness for the growing season was not sufficient for the LOMB site (Fig. 4). Timepoint interpolated data over the growing season was instead used for site comparisons (Fig. 5).

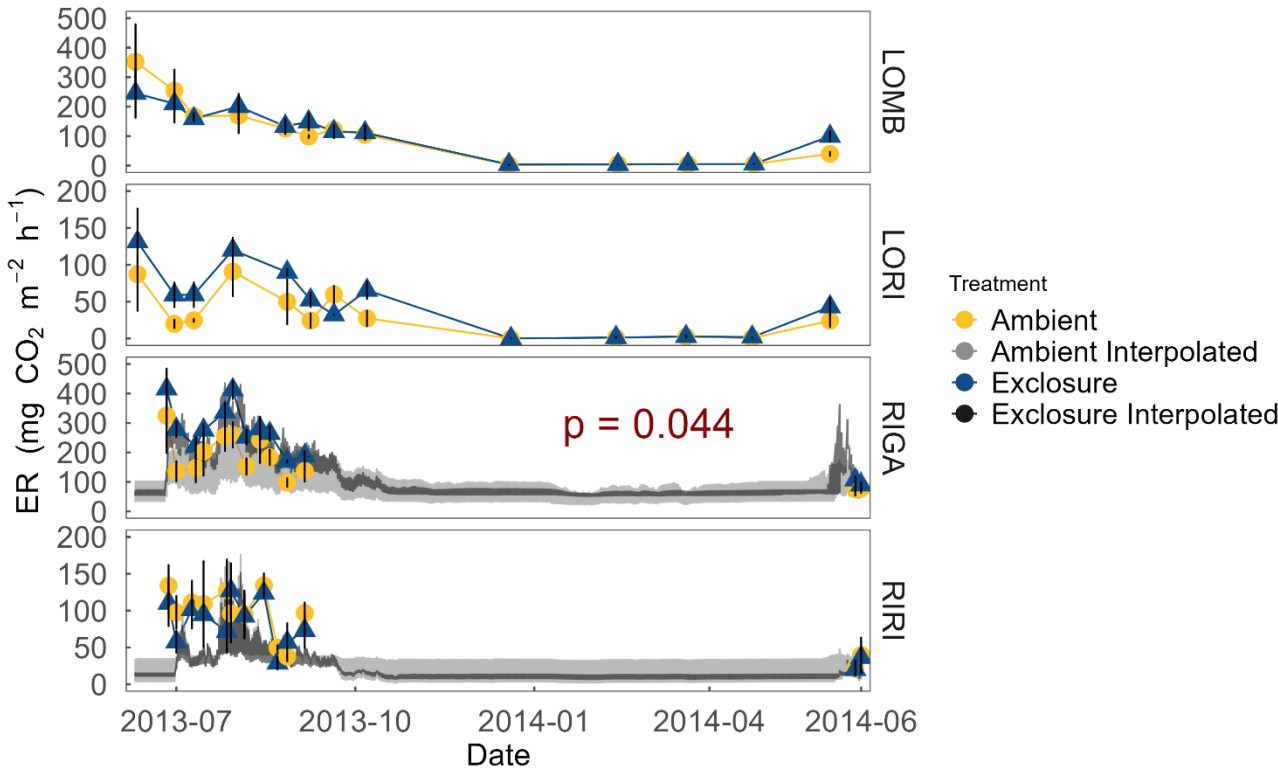

**Fig. 4.** ER at each site measured year-round from July 2013 to July 2014. ER was not measured over-winter in the Ritsem sites. Interpolated ER using associated $Q_{10}$-values is also plotted for sites with a significant Arrhenius relationship ($\alpha = 0.05$) as an estimate of ER on a shorter temporal scale. Exclosure and ambient treatments were not significantly different, except in RIGA where exclosure plots consistently had higher ER during the growing season. Error bars denote standard error of the mean.

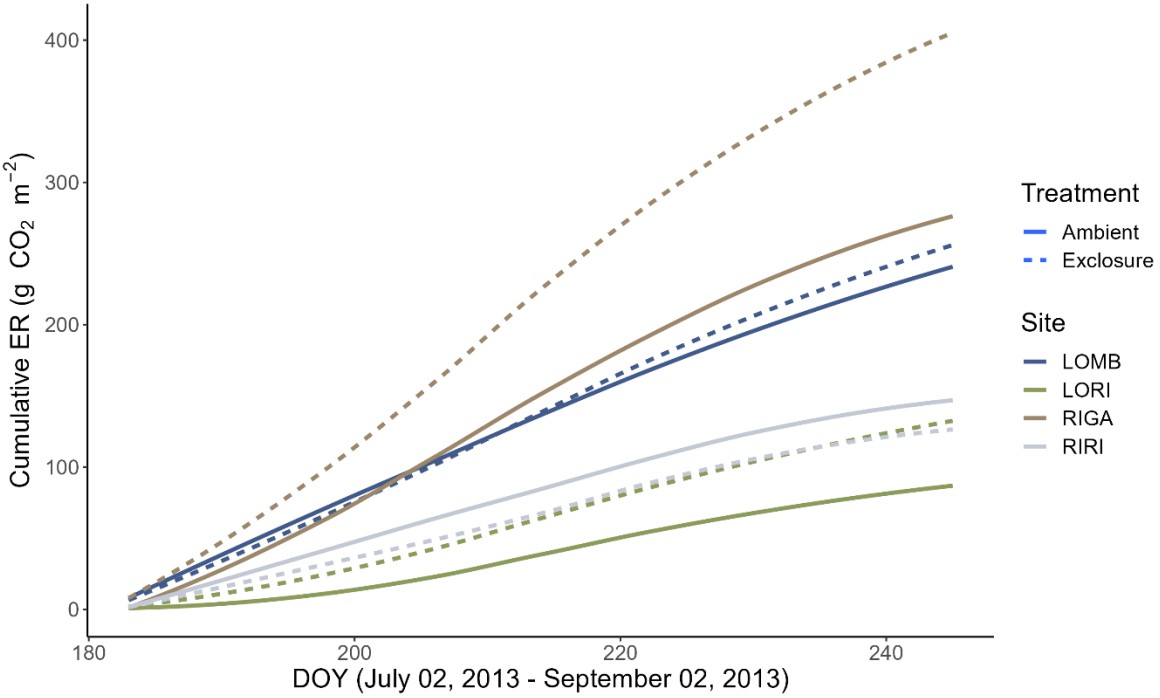

**Fig. 5. Cumulative ER between July 02 - September 02, 2013, in all four sites separated by treatment. The cumulative fluxes were calculated from interpolated fluxes that estimate the daily ER between measurement days using the slope and intercept values of the measured fluxes. Three of the sites had higher cumulative ER in the exclosure plots compared to control, with RIGA having the largest difference between the treatments.**

The overall linear mixed effects model indicated that ER was higher in the exclosure plots in RIGA over the growing season, whereas herbivory did not have a significant effect in the other sites (Fig. 4). Treatment ($F_{2,15}$ = 141.2, p < 0.001), site ($F_{3,15}$ = 21.3, p < 0.001), date ($F_{1,233}$ = 61.7, p < 0.001), soil temperature ($F_{1,233}$ = 5.8, p = 0.017), percent cover of graminoids ($F_{1,15}$ = 6.1, p = 0.026), and the interaction between soil temperature and graminoids ($F_{1,233}$ = 13.3, p < 0.001), were all significant predictors for the overall model. The RIGA-only mixed effects model did not find vegetation to be a significant predictor of ER, although evergreen prostrate dwarf shrubs were nearly significant ($F_{1,3}$ = 7.0, p < 0.077). However, treatment ($F_{2,3}$ = 74.3, p < 0.003), date ($F_{1,71}$ = 32.7, p < 0.001) and soil temperature ($F_{1,71}$ = 11.9, p < 0.001) were all significant predictors for the model.

### 3.3 Soil conditions

LOMB had the highest mean annual air temperature (2.6°C), followed by LORI (2.5°C), RIGA (0.5°C) and RIRI (-0.3°C), as measured by the site loggers (Fig. A2). TDD were significantly different between exclosure and

ambient plots at three of the four sites (LOMB: $t_{16} = 2.36$, p = 0.031, LORI: $t_{16} = 2.54$, p = 0.022, RIGA: $t_{16} = -3.16$, p = 0.006; Fig. A3); however, the direction of the treatment effect varied between sites. At the Långfjället sites, ambient plots were warmer than exclosure plots, while RIGA ambient were significantly cooler. No statistically significant differences in soil moisture were found between exclosure and ambient plots (Fig. A1). There were also no differences in the supply rate of $NO_3^-$ or $NH_4^+$ between any of the sites or the treatment conditions (Fig. A4). However, there was 3.5 times the amount of $NH_4^+$ in the southern sites (LOMB and LORI) compared to the northern sites (RIGA and RIRI) in winter; a pattern which was not discernible in the growing season.

## 4 Discussion

We found that BVOC composition was affected by large mammal herbivory, but to a lesser extent than by site, soil temperature and shrub dynamics. BVOC compositions in LOMB, LORI and RIRI were predominantly related to cover of different shrub types with RIGA not strongly tied to any specific shrub type. The interaction between herbivory and deciduous prostrate shrubs found in our study is likely a driver of the change in BVOC composition as long-term herbivore exclosure alters the vegetation at each site (Vowles et al., 2017a, b). The strong influence of plant functional type on BVOC composition agrees with previous research in Arctic environments (Männistö et al., 2023; Rinnan et al., 2020; Wester-Larsen et al., 2020), where shifts in graminoids and shrub species dominance can have a large effect on isoprene and terpene emissions. Previous research has also evaluated the effect of herbivory on BVOC emissions, but primarily from the perspective of insect herbivory or herbivory in general (Rieksta et al., 2021, 2020; Ghimire et al., 2021; Li et al., 2019; Faiola et al., 2015; Rinnan, 2013); explicit consideration of large mammal herbivory on BVOC dynamics is not well established (Vowles and Björk, 2019; Bartolome et al., 2007). Our results suggest that herbivores act as an indirect control on BVOC emissions via their influence on vegetation and soil temperature.

Two BVOC compounds were observed to have strong influence on distinguishing between individual BVOC measurements in terms of chemical composition. These compounds were β-pinene and 2-ethylfuran (the latter tentatively identified). Their emission rates generally increased toward LOMB and had a close trajectory to the treatment effect, indicating that they may be more common when herbivores are excluded. β-pinene is a monoterpene commonly produced by many species and is generally associated with defensive and antimicrobial properties, in particular it has been shown to affect bacteria, fungi and insect larvae (Silva et al., 2012; Mercier et al., 2009). The mixed-model indicated that β-pinene emissions tended to be lower in RIRI and compared to

LOMB, and therefore may be tied to the vegetation differences in the two sites. Since 2-ethylfuran was not quantifiable it is not possible to disentangle the strength of its vector from site or treatment effects. 2-ethylfuran is predominantly a plant defensive chemical with antimicrobial properties shown to be effective against fungi,

nematodes and can also inhibit seed germination in some species (Lazazzara et al., 2018; Aissani et al., 2015; Bradow and Connick, 1990). It has also been found to be emitted in BVOC samples from mountain birch and tundra ecosystems (Ryde et al., 2021; Wester-Larsen et al., 2020), although in low amounts. Since these compounds follow a similar trajectory as the treatment effect, it is possible that herbivory reduces the amount of these anti-microbial compounds emitted. This may be due to herbivory inadvertently selecting for vegetation that

emit these compounds, or vegetation producing more anti-microbial compounds when released from herbivory pressure. Alternatively, these compounds may serve a dual role as olfactory cues that function to improve foraging success in large mammals as the role of many volatiles in mammalian herbivory is not well known (Kimball et al., 2012; Moore et al., 2004; Palo, 1984). Further research into interactions between plants and large mammalian herbivores through BVOCs, especially regarding anti-browsing vs olfactory cues, may increase understanding of

herbivores' role in structuring plant communities, as our data suggest that large mammalian herbivory and specific BVOC compounds may influence each other.

Large mammal herbivory did not affect the magnitude of BVOC emissions, although there was a non-significant indication of reducing emissions. Contrastingly, insect herbivory has been shown repeatedly to increase BVOC emissions (Rieksta et al., 2021, 2020; Ghimire et al., 2021; Li et al., 2019; Faiola et al., 2015). However, studies

on direct insect herbivory usually evaluate the instantaneous impact on individual plants as opposed to the indirect effects of long-term vegetation shifts caused by large mammal herbivory. Direct herbivory effects are derived from the consumption of plant biomass typically without removing the entire individual but damaging the leaf tissue and result in a pulse of volatiles as the leaves or stems are broken and stimulates the production of herbivore deterrence linked compounds (Rinnan, 2013; Dicke, 2009). Large mammals may have similar effects on smaller

spatiotemporal scales, e.g., when consuming a single individual, but the effect of these herbivores on the composition of the vegetation community, especially shrubs, likely has a stronger indirect effect on the BVOC fluxes after 18 years of herbivore exclusion. The amount of monoterpenes emitted differed between sites with the southern sites generally having slightly higher emissions than the northern sites. The higher proportion of tall shrubs from the southern sites compared to prostrate shrubs in the north may drive these differences in MT

emission (Ghirardo et al., 2020). Contrastingly, the amount of sesquiterpenes did not change across any of the sites and was much lower than the MT emissions in the southern sites. The change in MT and SQT emissions between sites is related to the dominant vegetation at each location where shrub species in simulated tundra

ecosystems have been shown to contribute to terpene formation to different degrees (Ghirardo et al., 2020). BVOC emissions vary substantially between individual plant species and so direct patterns can be difficult to detect in a community (Rinnan et al., 2011). Emissions also vary seasonally and with temperature (Rinnan et al., 2011; Faubert et al., 2010), and environments can take up a portion of VOCs from the atmosphere (Baggesen et al., 2021), which was not evaluated in this study. Herbivory effects did not provide a strong enough signal in the magnitude of BVOCs emitted to determine a change compared to the high amount of variation in these compounds naturally.

Herbivory by large mammals significantly decreased ER in RIGA, but no effect was found in the other three communities. RIGA also has the highest plant productivity (Vowles et al., 2017b), which results in this community having larger ER in general, as shown elsewhere (Treat et al., 2018). By influencing vegetation composition herbivores should alter ER; however, no interaction was found between the treatment and any plant functional type examined despite RIGA exclosure plots having a large increase in tall deciduous shrubs compared to semi-prostrate evergreen shrubs in ambient plots (Vowles et al., 2017a).

Soil temperature was the primary driver of the difference between the treatment conditions, suggesting that herbivory indirectly affects soil temperature through trampling and/or modifying the vegetation community, or by modifying microbial communities. Trees and tall-statured shrubs have been found to increase soil temperatures in the Arctic (Kropp et al., 2021), however, there may still be a cooling effect of shading in the immediate microclimate. It could be that increases in tall woody vegetation causes the initial thaw to happen earlier or occur quicker, and even low statured shrub vegetation had higher maximum soil temperatures than graminoid tundra (Kropp et al., 2021). The observed changes in vegetation at RIGA due to the exclosures could explain part of the observed soil temperature effect on ER, without these vegetation changes necessarily being strong enough to have a significant impact on ER themselves. ER may also increase under warmed conditions regardless of vegetation community (Ward et al., 2013), which suggest that the microbial community may be more important. Changes in the microbial community due to the temperature differences may also drive the change in ER as belowground communities are responsible for a large portion of ER (García et al., 2023) and their temperature adaptation is influenced by maximum summer temperature (Rijkers et al., 2023). Higher temperatures relate to higher microbial activity which can increase ER. Johnston et al. (2021) calculated a breakpoint in the relationship between soil temperature and ER globally at 11.4 °C $\pm$ 0.29 °C, after which increases in temperature have a reduced effect on ER, which is very near the observed soil temperature difference between treatments at RIGA (11.6 °C $\pm$ 0.61 °C in exclosure plots and 10.2 °C $\pm$ 0.65 °C in ambient plots). Part of the shift in ER observed could be due to the

relatively small difference in soil temperature between the treatment conditions as they occur below to just above this critical threshold.

RIGA has the largest proportion of graminoid vegetation which is a significant predictor of ER between the sites. Our results suggest that the interaction between graminoids and soil temperature accounts for much of the between site differences. The interplay between these properties and shrub species in a single community may have a strong effect on ER as both have been found to be important drivers of $CO_2$ emissions (Ward et al., 2013; Cahoon et al., 2012). The direction of the herbivory effect, although not always significant, was consistent across the sites,

except at RIRI. However, the vegetation changes due to herbivory exclusion within LOMB, LORI and RIRI were not consistent, with LOMB showing an increase in evergreen shrubs and a reduction in graminoids and the heath sites primarily increasing in deciduous shrubs (Vowles et al., 2017b). The lack of significance in LOMB could be tied to its lack of change in deciduous shrubs which can have a disproportionate effect on soil respiration (Cahoon et al., 2012), or a lack of change in soil temperature between the treatment conditions, as that is what drove the

differences in RIGA. The non-significant herbivory effect on ER in our heath communities may be due the low productivity of these communities leading to low ER (Min et al., 2021; Liu et al., 2018; Treat et al., 2018); therefore, the effect size may be too small to be discerned statistically.

    The overall pattern of increased ER when herbivory is excluded agrees with changes observed in a wet tundra site, where long-term removal of lemming herbivory increased the net production of $CO_2$ by reducing productivity

(Lara et al., 2017). In this case, when lemmings were excluded, the habitat shifted from one dominated by graminoids to a less productive community with higher proportions of mosses and lichen. However, the increase in $CO_2$ emission was only observed in net ecosystem exchange (NEE). In fact, few studies have found a significant effect of herbivory on ER specifically, even if those studies did find links between herbivory and NEE (Du et al., 2022; Hu et al., 2017; Lara et al., 2017); although, there has been a link shown between herbivory and ER via soil

temperature in a high arctic wet meadow site grazed by geese (Sjögersten et al., 2011). The findings of our study are unique as it suggests that the herbivores affect ER in a high productive meadow habitat, but not in low productive heaths. Thus, herbivory can act to modify the magnitude of fluxes in tundra communities with the root determinant of these fluxes being the characteristics of the community itself (e.g. moisture, dominant vegetation, etc).

## 5 Conclusions

Our data suggest that herbivory alters BVOC emission composition through its effects on the proportion of shrub species, however there is no effect on the magnitude of these emissions. Large mammal herbivory also alters the strength of ER in a productive meadow community but is not strong enough to have an impact in heath and mountain birch ecosystems. The influence of large mammals on graminoid and shrub dynamics likely drives the effects observed as these species have a disproportionate effect on the ecosystem's productivity and carbon fluxes. Predicted changes in plant communities, especially shrub encroachment, will feedback onto climate change through their effect on carbon fluxes and these effects may be situationally mediated by herbivory. Overall, the productivity of plant communities and the capacity for soils to support plant growth are likely most important for climate change feedbacks in tundra ecosystems.

## Appendix A

**Table A1. Terpene emissions from each site averaged for each sampling date. Sites were measured twice in summer of 2013, once near the start of the growing season and once near the end. Compounds were identified through the NIST database and quantified by comparing to pure standards. All units are ng m$^{-2}$ h$^{-1}$, standard error is displayed in brackets.**

| Site | Sampling date | Total terpene emissions | Monoterpene emissions | Sesquiterpene emissions |
|------|---------------|------------------------|----------------------|------------------------|
| LOMB | 2013-07-05 | 418.3 (98.4) | 880.0 (207.9) | 49.0 (10.8) |
| | 2013-08-01 | 193.5 (46.4) | 416.6 (101.7) | 15.1 (2.2) |
| LORI | 2013-07-06 | 248.3 (54.9) | 423.4 (74.5) | 108.1 (39.2) |
| | 2013-08-02 | 129.1 (23.1) | 242.6 (30.1) | 38.3 (17.4) |
| RIGA | 2013-07-11 | 64.8 (7.7) | 137.6 (16.2) | 6.5 (0.8) |
| | 2013-07-26 | 113.3 (14.4) | 236.1 (26.6) | 15.1 (4.6) |
| RIRI | 2013-07-10 | 41.9 (3.6) | 84.3 (6.5) | 8.0 (1.3) |
| | 2013-07-27 | 117.6 (23.3) | 186.2 (20.1) | 62.8 (25.8) |

**Table A2. Mean terpene emissions from each site and treatment condition. Standard error is reported in brackets. Compounds were identified through the NIST database and quantified by comparing to pure standards. All units are ng m$^{-2}$ h$^{-1}$.**

| Site | Treatment | Unidentified MT1 | Unidentified MT2 | α-pinene | β-pinene |
|------|-----------|------------------|------------------|----------|----------|
| LOMB | Exclosure | 91.5 (45.8) | 209.8 (75.9) | 1764.7 (689.2) | 2664.3 (356.7) |
|      | Ambient   | 39.2 (5.3) | 117.0 (35.4) | 846.6 (441.9) | 2346.4 (352.8) |
| LORI | Exclosure | 32.2 (1.3) | 48.3 (3.8) | 196.2 (48.9) | 1782.0 (213.4) |
|      | Ambient   | 32.1 (0.7) | 46.1 (2.4) | 148.4 (22.4) | 1523.4 (128.6) |
| RIGA | Exclosure | 29.8 (0.9) | 41.8 (3.7) | 81.2 (8.4) | 1178.2 (143.0) |
|      | Ambient   | 29.5 (1.1) | 39.7 (3.3) | 124.5 (28.3) | 931.2 (119.0) |
| RIRI | Exclosure | 29.5 (1.3) | 40.5 (2.6) | 83.0 (11.2) | 722.0 (152.8) |
|      | Ambient   | 29.7 (0.8) | 38.5 (2.0) | 71.0 (5.7) | 651.0 (92.0) |

| Site | Treatment | 3-Carene | Cymene | D-Limonene | Linalool |
|------|-----------|----------|--------|------------|----------|
| LOMB | Exclosure | 353.7 (120.9) | 550.2 (270.8) | 200.1 (74.4) | 206.2 (39.1) |
|      | Ambient   | 401.0 (157.8) | 269.5 (86.8) | 147.2 (48.7) | 166.1 (28.6) |
| LORI | Exclosure | 59.4 (5.9) | 121.9 (21.6) | 64.4 (6.8) | 753.1 (391.9) |
|      | Ambient   | 84.8 (10.8) | 96.3 (9.5) | 55.4 (3.9) | 340.2 (78.9) |
| RIGA | Exclosure | 32.7 (3.2) | 61.2 (4.9) | 49.9 (6.3) | 82.1 (14.0) |
|      | Ambient   | 68.4 (18.0) | 105.0 (29.5) | 58.2 (10.9) | 76.8 (5.6) |
| RIRI | Exclosure | 37.8 (4.7) | 77.8 (11.6) | 40.8 (3.9) | 110.9 (23.3) |
|      | Ambient   | 34.8 (2.0) | 67.5 (4.0) | 37.8 (2.2) | 100.5 (17.6) |

| Site | Treatment | α-Cubebene | Unidentified SQT1 | Unidentified SQT2 | Unidentified SQT3 |
|------|-----------|------------|-------------------|-------------------|-------------------|
| LOMB | Exclosure | 18.3 (7.0) | 31.3 (15.1) | 28.0 (8.2) | 27.3 (10.1) |
|      | Ambient   | 18.9 (5.9) | 14.9 (2.1) | 19.0 (2.1) | 27.9 (8.6) |
| LORI | Exclosure | 98.3 (47.6) | 17.2 (3.4) | 22.5 (4.4) | 112.3 (56.8) |
|      | Ambient   | 37.7 (13.5) | 13.2 (1.2) | 24.6 (8.2) | 39.1 (14.2) |
| RIGA | Exclosure | 3.4 (0.5) | 13.5 (1.9) | 13.3 (3.0) | 17.6 (11.0) |
|      | Ambient   | 3.4 (0.4) | 10.9 (1.7) | 16.8 (8.4) | 7.8 (3.7) |
| RIRI | Exclosure | 13.8 (7.2) | 23.9 (8.1) | 29.7 (14.4) | 23.2 (9.8) |
|      | Ambient   | 14.5 (6.3) | 82.3 (64.8) | 42.0 (22.0) | 17.8 (6.1) |

| Site | Treatment | Germacrene D | α-Bourbonene | Copaene | Caryophyllene |
|------|-----------|--------------|--------------|---------|---------------|
| LOMB | Exclosure | 34.2 (13.9) | 33.8 (10.6) | 89.0 (27.5) | 46.4 (8.0) |
|      | Ambient   | 35.6 (11.8) | 28.9 (9.2) | 103.1 (28.5) | 39.5 (7.3) |
| LORI | Exclosure | 143.6 (69.0) | 59.9 (21.7) | 349.1 (138.3) | 128.5 (53.2) |
|      | Ambient   | 56.8 (19.9) | 32.8 (9.9) | 158.5 (52.4) | 67.5 (19.4) |
| RIGA | Exclosure | 4.3 (0.9) | 10.1 (1.8) | 10.7 (3.0) | 32.5 (10.0) |
|      | Ambient   | 3.8 (0.5) | 6.1 (1.2) | 6.3 (0.7) | 14.4 (2.6) |
| RIRI | Exclosure | 24.7 (12.0) | 41.9 (23.9) | 103.2 (51.1) | 17.1 (6.9) |
|      | Ambient   | 24.5 (10.0) | 90.5 (54.7) | 95.7 (46.6) | 16.3 (4.8) |

| Site | Treatment | α-Farnesene | cis-Calamenene |
|------|-----------|-------------|----------------|
| LOMB | Exclosure | 10.2 (1.1) | 12.1 (2.3) |
|      | Ambient   | 9.9 (1.0) | 12.0 (2.1) |
| LORI | Exclosure | 73.3 (30.8) | 33.1 (15.1) |
|      | Ambient   | 25.4 (8.0) | 14.8 (4.1) |
| RIGA | Exclosure | 12.1 (3.5) | 12.3 (5.9) |
|      | Ambient   | 9.5 (2.4) | 6.2 (1.4) |
| RIRI | Exclosure | 11.5 (4.6) | 9.4 (3.2) |
|      | Ambient   | 8.5 (1.5) | 9.2 (2.2) |

**Table A3. Air temperature recorded for each sampling date at each site.**

| Site | Sampling date | Air Temperature (°C) |
|------|---------------|----------------------|
| LOMB | 2013-07-05 | 18.8 |
|      | 2013-08-01 | 15.9 |
| LORI | 2013-07-06 | 19.1 |
|      | 2013-08-02 | 20.5 |
| RIGA | 2013-07-11 | 8.3 |
|      | 2013-07-26 | 22.4 |
| RIRI | 2013-07-10 | 6.2 |
|      | 2013-07-27 | 22.4 |

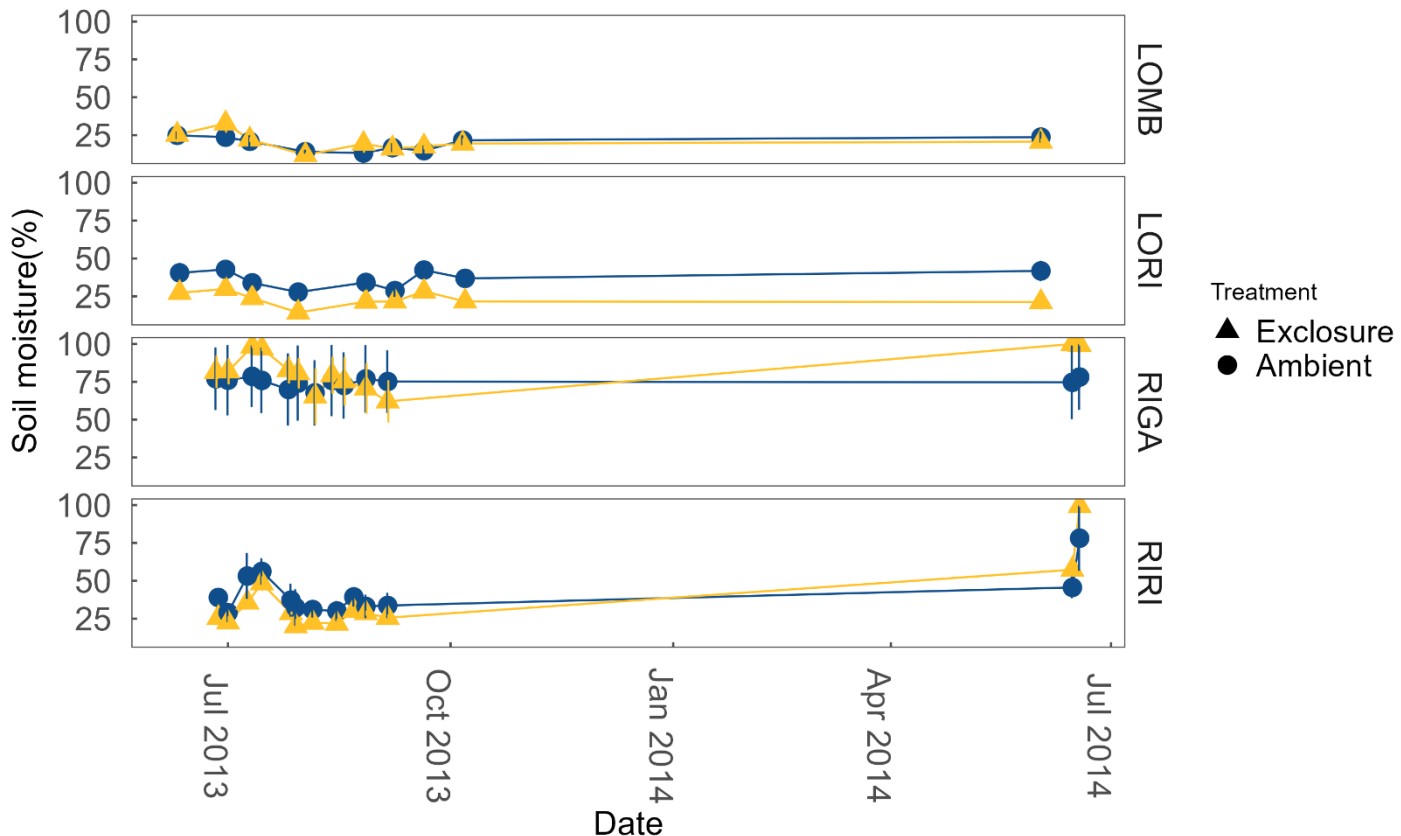

**Fig. A1. Soil moisture at each site and treatment condition over the full year.**

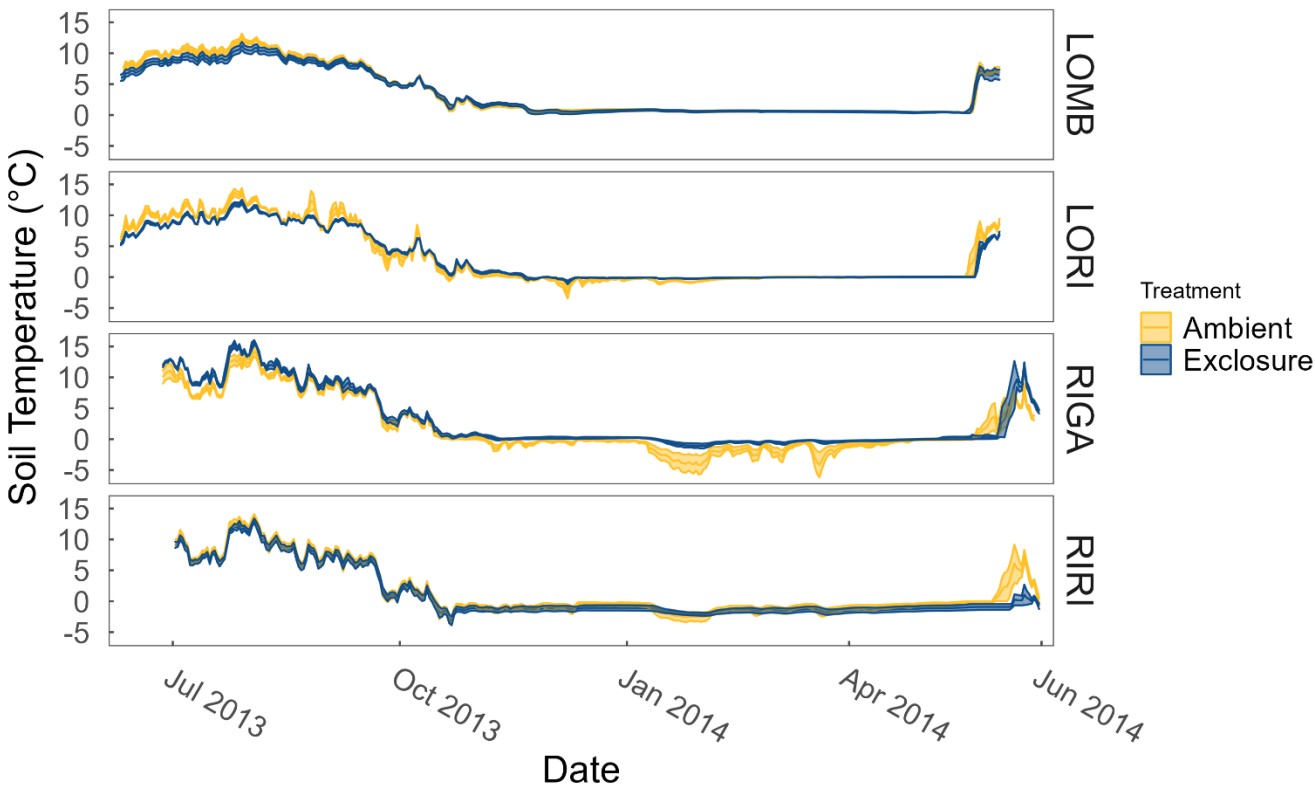

**Fig. A2.** Soil temperatures at each site and treatment measured year-round by temperature loggers (Tinytag plus 2 TGP-4020; Gemini Data Loggers, Chichester, UK) at 2 cm depth. Ribbons correspond to the standard error around the mean.

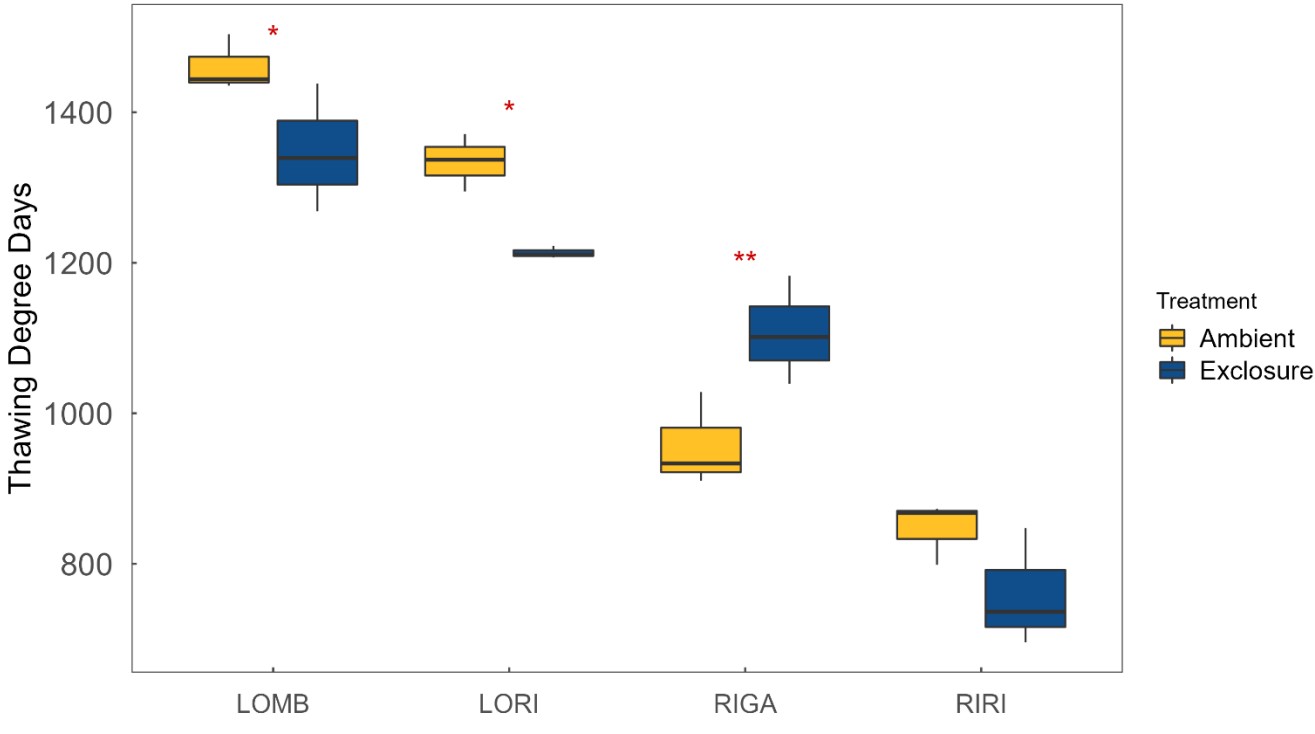

**Fig. A3. Thawing degree days, measured as the sum of all mean daily temperatures above 0°C calculated from the soil temperature data, for each site and treatment. Significant differences between ambient and exclosure conditions were found for LOMB, LORI and RIGA, with ambient having more TDD in the southern sites (LOMB and LORI) and fewer in RIGA.**

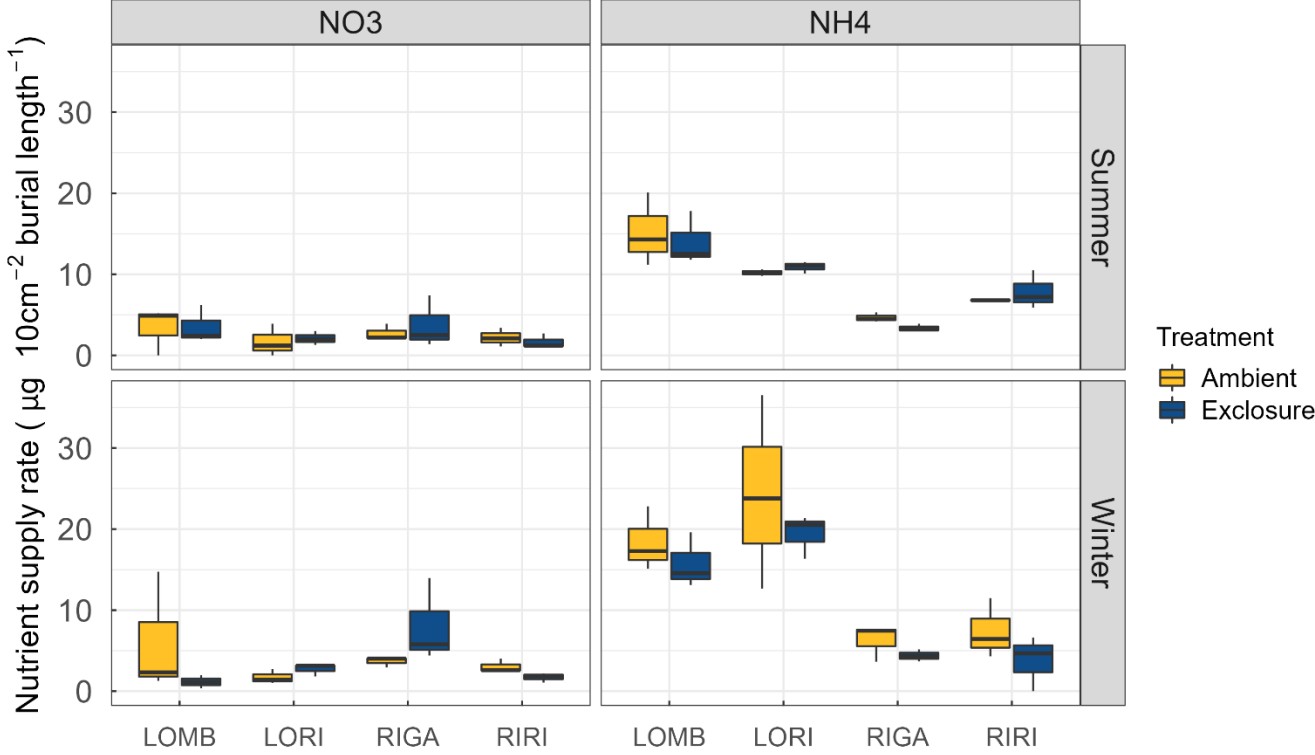

**Fig. A4. Supply rate of NO$_3^-$ and NH$_4^+$ in each site and treatment condition. The supply rate is calculated for the surface area of the probe (10 cm$^2$) and the burial length. The burial length was different for southern and northern sites and varied between seasons. LOMB and LORI probes were buried for 121 days in summer and 235/236 in winter; respectively. RIRI and RIGA probes were buried 71 days in summer and 283 in winter. Nitrogen was measured through PRS probes and transformed to supply rate following**
**manufacturers protocol. No significant differences were found between exclosure and ambient conditions for any site or nutrient. There was, however, a significant decrease in the amount of NH$_4^+$ in the northern sites compared to the southern sites in winter.**

## Author contributions

RGB and TV designed the experiments. RGB, TV and MPB led the field campaigns. RR and TV processed the raw data. CGB

analyzed the data and wrote the original manuscript. All coauthors provided edits and reviewed the manuscript.

**Competing interests**

The authors declare that none of the authors has any competing interests.

## Acknowledgement

We would like to thank our dedicated field assistants, Kjell Vowles, Lars Lindstein, Oscar Lidskog, Martina Jonsson and Erika Jonsson for their work collecting the data for this paper. We would also like to thank the reviewers who have helped improve this manuscript through the submission process. We would like to acknowledge Formas (grant no 214-2010-1411 to RGB), the Swedish Research Council (grant no 2018-04202 to RGB), the European Research Council (ERC) under European Union's Horizon 2020 research and innovation program (grant no 771012 to RR), and the Elite Research Prize of the Danish Ministry for Higher Education and Science (grant no 9095-00004 to RR) for their funding contributions. The research presented in this paper is a contribution to the strategic research area Biodiversity and Ecosystems in a Changing Climate, BECC.

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
