# Peer review of "Herbivore-shrub interactions influence ecosystem respiration and BVOC composition in the subarctic"

_Biogeosciences, 2023_

## Author Response (AR1)

Response to associate editor:

Dear Trisha Atwood,

Thank you for the opportunity to resubmit following major revisions. We have updated the manuscript according to the reviewers' comments and believe we have addressed the identified issues and strengthened the paper. The most substantial update has been to the ER model which has some effect on the results and larger consequences for the discussion section adding in more nuance to the overall take-away of how herbivores can affect ER via the vegetation community. We appreciate your feedback and your help in improving this manuscript greatly from first submission!

Sincerely,

Cole Brachmann (on behalf of the coauthors)

Responses to reviewer comments:

Dear Reviewer 1,

Thank you again for your detailed and helpful comments. We have addressed each in the revised manuscript with detailed changes described below. Most of the changes are in the methods and corresponding parts of the results and discussion sections. The most substantial update has been to the ER model which influences the results and discussion sections. We appreciate your feedback and your help in improving this manuscript greatly from first submission!

Sincerely,

Cole Brachmann (on behalf of the coauthors)

We have updated the linear mixed-model for the ER to include percent cover of graminoids as they had a significant contribution to the model mainly explaining some of the site level differences. We have then updated the results and discussion sections accordingly. More detailed descriptions of changes made can be found below:

**Line 43-46: This sentence is really the only one in the introduction that explains the importance of BVOCs to climate change. Arguably, the results of this study related to BVOCs are the most robust and interesting aspect of the study, but most readers will not be very familiar with their importance to climate change. Could the authors provide additional information about the relevance of BVOCs to climate change?** Additional background information detailing the importance of BVOCs on climate change has been added to the introduction section (Lines 39-41, 44-46, and 57-63). To further address this point and others below, we have restructured the order in which BVOCs and ER are discussed throughout the manuscript to put BVOCs first and ER second.

**Line 75-77: Why will excluding herbivores shift plant dominance to deciduous shrubs? The introduction does not provide specific background about why this should occur and should be revised to include this information.** Information has been added about the previous observed vegetation changes at the study sites to rationalize why it was believed that deciduous shrubs would respond most strongly to herbivory (Lines 89-90).

**Line 77-79: This hypothesis does not reflect your statistical approach. As stated in the general comments section above, plant functional type is not included in your statistical model for ER, and soil temperature is the only soil characteristic considered. Furthermore, it doesn't appear that climate properties were included in any models for either ER or BVOCs.** The model for ER has been updated upon more thorough testing with vegetation data. The model now includes graminoids as a significant predictor, which has shifted the story to being about the interplay between shrubs and graminoids and how herbivores effect these groups in different communities. The role of graminoids mostly explains the site-level differences with the herbivory effect in RIGA being primarily driven by soil temperature differences (see Lines 391-409 for discussion). We believe the improvements made to the statistics and the rewording of the hypotheses as a result has improved the accuracy of the hypotheses. Soil properties (soil temperature, soil moisture, available $NH_4^+$ and $NO_3^-$) and climate variables (air temperature) were included in the hypotheses as they were tested in the models, however only soil temperature was found to improve the models and so was the only soil predictor included.

**Line 95: What does it mean for a density estimate to be "tentative"?** The word "tentative" was not suitable for the sentence and so was removed and the sentence reworked to reflect the change. The sentence now reads "The primary large mammalian herbivore in our sites is reindeer (*Rangifer tarandus tarandus*) for which density was estimated as 2.8 reindeer per km2 near Långfjället and 2.2 reindeer per km2 near RIGA and 1.4 reindeer per km2 near RIRI previously reported for the three Sami herding villages nearest our sites (Vowles et al., 2017b, a)."

**Line 97-98: The sentence beginning "Langfjallet is an area…" is unnecessary because the previous sentence already reports reindeer density, and the proximity of RIGA and RIRI is reported earlier in the paragraph.** The sentence was originally included to highlight the differences in reindeer density between the sites, however, we agree that it is unnecessary and has been removed.

**Lines 106-121, Table 1: It is unnecessary to report climate conditions both in the paragraph and as a table. I would recommend keeping the table and eliminating the climate description from the paragraph.** We agree that having both description and table are redundant and so we removed the description and direct to the table instead.

**Lines 128-131: How many pairwise plots were present at each site? It seems as if there are three at most sites, and two at RIGA. Is this correct? Please explicitly state the number of plots at each site.** The wording of this sentence has been improved to explicitly mention the number of paired fences and ambient plots for each site (which is three of each per site) as the amount and distinction between sites was not included previously. One ambient plot was lost in RIGA and so a new ambient plot was established in 2012 to keep the number of paired treatment-ambient plots equal. The sentence now reads "The effect of herbivory on ER and BVOC fluxes was determined using herbivore exclosure fences. Three fences and three paired ambient plots (25 x 25 m) were installed at each site in 1995 and are composed of wire mesh 1.7 m high that functions to exclude reindeer and other large mammalian herbivores from accessing the sites (Vowles et al., 2017b, a)."

**Line 221-222: This sentence seems to be missing a word, making it difficult to interpret. However, I don't think this justification is necessary as using a linear mixed model for categorical data is not an unusual approach.** This sentence has been removed following your comment as we agree that the extra justification is unnecessary.

**Line 226-229: This sentence appears to be missing some punctuation that makes it difficult to read.** A comma was added after "As a constrained ordination…" to improve clarity of the sentence and address your comment.

**Line 235: There should be a comma after "sites".** A comma has been added.

**Line 239: I recommend changing "fences" to "fenced" here and throughout.** We have changed fences to fenced where appropriate within the document.

**Lines 305– 315: Here, the authors attempt to link their findings of differences in ER to plant community composition, but, as stated in the general comments, these claims are not well supported statistically by the authors' models. The authors should either explicitly include plant composition in their statistical approach, or consider additional explanations for the observed patterns in ER. For example, the authors found that temperature was a significant predictor of ER, and RIGA, the only site with differences in ER, is also the only site where soil temperature in ambient plots is consistently lower than in exclosure plots. Doesn't this suggest that temperature is more likely than plant composition to explain these patterns? Why is the influence of temperature not discussed?** We have amended the models to include graminoids as a significant predictor of ER between sites. Additionally, the role of soil temperature is now discussed more thoroughly in the discussion section (Lines 391-409) as the main driver between treatments is related to soil temperature effects.

**Lines 305 – 331: This is a long paragraph that is difficult to follow because it jumps from topic to topic in a fairly rapid fashion. I would recommend restructuring this paragraph to keep related topics together, and perhaps consider breaking it up into two separate paragraphs.** We agree that the paragraph is long and contains two separate ideas which we have split into two separate paragraphs (Lines 327-361 for these two paragraphs).

Dear Dr. Kelsey,

Thank you again for your insightful comments. We have addressed each in the revised manuscript with detailed changes described below. Your comments have helped with clarity and readability of the manuscript and are greatly appreciated.

Sincerely,

Cole Brachmann (on behalf of the coauthors)

**How many herbivory exclosures were present at each site? Three? I tried to find this information in the methods but it was difficult to tell.** Information on the layout of the fences and the number of plots were added to the methods section (Lines 136-139).

**While the exact sampling dates are recorded in the supplemental material, but a brief description of the dates (i.e. early July and early August) would be appropriate to include in the Methods.** We have added the approximate time of year for the $CO_2$ measurements (Lines 170-171), and for BVOC measurements (Line 145).

**The methods say that vegetation was measured in each "plot" – does this mean each exclosure? How many "plots" at a "site"?** Yes, this refers to each exclosure and each ambient plot, these plots are defined near line 136 as each fence and ambient sampling area (25 x 25 m). There are 6 plots per site as there are three fences and three ambient plots at each site. For the vegetation assessments, 20 1 $m^2$ subplots were used within the larger plots (excluding edge area).

**Was plot included within the linear mixed effects models?** Plot was included as a random factor (effect) in the linear mixed effects models (Lines 222 and 236).

**Line 240 what is meant by "but not with growing season data alone"? This is unclear.** This was originally included to make clear that these relationships were not consistent when analyzing just growing season data, but rather relied on the overwinter fluxes which were only available for the Långfjället sites. Upon rereading it, we agree that it is not necessary and so was removed.

**On some occasions the vegetation communities are referred to using a four-letter code, and in some cases by a description (e.g. low herb community). Personally, I find the description much easier to read, but whichever the authors choose should be consistent throughout the manuscript.** We have changed all instances of referring to one of our sites with the acronym assigned, but still use community type when referring to it in a general context. I hope this maintains consistency while easing communication in the discussion section.

**Line 320 - I recommend also consulting Leffler et al., 2018, Kelsey et al., 2018, Sjogersten et al, 2011 regarding the effects of herbivory on ecosystem respiration.** Thank you for the suggested papers. We have added the Sjögersten et al 2011 paper (Lines 429-430), but have decided against the other two as we felt their focus on the phenological mismatch of herbivory was outside the scope of this paper.

Technical corrections:

**Line 305 – suggest changing to "no effect was found in the other three communities." There are other places in the text where a similar change would improve the clarity.** The suggested update to the text was included and further areas that could also benefit from the change were updated.

**Figure 2 – the lables of "Ambient Q10" and "Exclosure Q10" are somewhat misleading – it appears at first glance that the figure is reporting Q10 values for this site, but rather this is the method used to produce the interpolated data. It would be more clear to label these lines with "… interpolated" or something similar.** The labels have been updated to "Ambient interpolated" and "Exclosure interpolated".

**Figure 2 and 3 – one figure uses the term "Exclosure" whereas the other uses the term "Fenced". These should be consistent – the authors may want to consider using the terms "grazed" and "ungrazed" as this describes the treatments more specifically. But, using the term "exclosed" or**

**"fenced" is also fine, as long as it is consistent.** Figure 3 was updated to also use "exclosure" instead of "fence".

**Figure 4 – in the caption it states "letters denote significant differences" but there are no letters in the Figure.  I believe there is not significant difference in these data, so all bars should have the same letter, or this should be removed to limit confusion.** The letters were not copied properly onto the figure. It has been adjusted now, in general there was no difference except between both RIRI conditions and LOMB exclosure in terms of magnitude of monoterpene emissions.

---

## Author Response (AR2)

Dear Dr. Atwood,

Thank you for your suggested improvements to the manuscript they help greatly in improving readability and clarity in the paper. We have accepted many of your suggestions and took careful consideration of those we chose not to include. Below is a detailed response to your comments we hope addresses the concerns. We appreciate your input and believe the manuscript is overall improved.

Sincerely,

Cole (on behalf of the coauthors)

Detailed responses:

The suggested grammar changes have been incorporated in the manuscript; we thank you for pointing out where the writing could have been improved.

**The use of "plot" and "exclosure" is still confusing. If you want to call them all plots, add something along the lines of "hereon referred to as plots" to line 136.** I understand the potential confusion, to help improve clarity they are now referred to as "exclosure plot" and "ambient plot" throughout the manuscript instead of exclosure/fence and ambient plot.

**Line 220: What is meant by treatment here (and throughout)? Is this the vegetation type or +/- herbivore?** Treatment in this context and in the models refers to whether herbivores are present or excluded as it is the experimental change made to the system. In most instances treatment is just denoting that there is a difference between the conditions, with the direction of change being stated directly where appropriate. I have added a sentence to the BVOC composition results section that has made the direction of the vector in Figure 5 more explicit to help with clarity (Line 274-275).

**In the statistical approach section: It would be helpful to identify which of your objectives is being tested by which statistical approach. To examine the effects of herbivore-induced changes in vegetation type on BVOC composition and magnitude, we used…..** This is a great suggestion; I have added a lead-in to each sentence before a statistical test to connect it back to the main aims of the study. (Lines 220, 228, and 237)

**Line 243: what are the new Bonferroni P-values?** The p-values listed from the mixed effects models for ER are the Bonferroni adjusted p-values as they adjust for multiple comparisons to reduce the family-wise error rate. We do not list the non-adjusted values as they are less accurate when looking at comparisons between multiple groups.

**Line 244: Again, what is meant by treatment? Is this only referring to herbivory because RIGA only included one type of vegetation?** Yes, treatment in the manuscript is only referring to the use of herbivory fences to exclude large mammals and not to the presence of different vegetation types or soil properties as they were not altered directly but rather as a consequence of the fences and

differences in environmental conditions between sites. I hope the addition of "…treatment (herbivory present in ambient plots or absent in exclosure plots)…" to Line 220 helps to clarify our intention.

**Line 266: what is % cover of abiotic components?** Percent cover of abiotic components is a catch-all term we used to refer to number of hits from the point frame that miss vegetation entirely and land on stone, soil, woody debris, etc; really anything that is not vegetation during the point framing.

**Line 275: What is meant by tentative NIST identification?** The identification of compounds using the NIST spectral library is an estimation of what the compound is based on how well its spectra matches different compounds in the database. Due to potential issues such as the database not having a good (or any) reference for a given sample one tries to identify; we chose to phrase the identifications as tentative instead of definitive as would be determined by comparing to a known reference standard instead of a database.

**In the discussion and conclusion: It would be helpful to clearly identify which BVOCs were different between herbivore present and herbivore absent plots. It would then be helpful to discuss if this difference means anything for climate change or other processes. In other words, does the observed difference in BVOC composition matter?** We appreciate the suggestion and do agree that it would be nice if the importance and implications of the observed BVOC composition differences could be discussed more. However, we prefer not to add discussion on this any further than already present in the manuscript regarding the significant vectors in the RDA, beta-pinene and 2-ethylfuran. The reason is that we understand poorly the effects of relative changes in the BVOC composition on atmospheric reactivity and composition, so adding anything would be rather vague and speculative.